

# Reply to Lars Olof Björn's Comment on our article "Fundamental molecules of life are pigments which arose and co-evolved as a response to the thermodynamic imperative of dissipating the prevailing solar spectrum"

Karo Michaelian[1] and Aleksandar Simeonov[2]

[1]Department of Nuclear Physics and Application of Radiation, Instituto de Física, Universidad Nacional Autónoma de México, Circuito Interior de la Investigación Científica, Cuidad Universitaria, Cuidad de México, C.P. 04510
[2]Faculty of Natural Sciences and Mathematics, Ss. Cyril and Methodius University in Skopje, North Macedonia

**Correspondence:** Karo Michaelian (karo@fisica.unam.mx) and Aleksandar Simeonov (alecsime.gm@gmail.com)

**Abstract.** Lars Björn, in his critique of our article, doubts our assertion that at the origin of life the fundamental molecules of life (those in the three domains) were UVC pigments, dissipatively structured under a thermodynamic imperative to absorb and dissipate this UVC light into heat. Björn bases his critique on the suggestion that non-living material can be more photon absorbing than living material. He gives a number of examples in which he shows that the albedo of material devoid of life is lower than that of biotic material and concludes that these examples counter our assertion. However, Björn makes the erroneous assumption that albedo (reflection) is the only important factor related to photon dissipation (entropy production) occurring in the light-pigment interaction in living systems. He ignores the other contributions to entropy production due to the photon interaction which were listed in our article; 1) the shift towards the infrared of the emitted spectrum, 2) the diffuse emission and reflection of light into a greater outgoing solid angle, 3) the coupling of photon-induced evapotranspiration in the pigmented leaf to further photon dissipating processes such as the water cycle, which further allows dissipating biopigments to flourish over all of Earth's surface. His assumption is therefore incorrect and his analysis does not provide legitimate reason for doubting our assertion that the fundamental molecules of life arose as pigments as a response to the thermodynamic imperative of dissipating the prevailing solar spectrum.

In the following, we respond to each critique using the same section headings of Björn's Comment.

## 1  Introduction: Do living systems reduce the albedo of Earth?

Contrary to Björn's assertions, living organisms do, in fact, generally reduce the albedo with respect to regions devoid of life, and there are many data available in the literature. For example, the visible albedo of deciduous forests is 0.15 to 0.18 and that



of coniferous forests is 0.09 to 0.15, while that of sandy deserts is about 0.30 [Barry and Chorley (1992)] and rocky deserts (Gobi) 0.21 [Wang et al. (1998)]. And this is also true at wavelengths greater than the red-edge (∼ 700 nm), for example, forest albedo increases to about 0.3 [Coakley (2003)], while sand and rocky desert albedo increases to about 0.50 [Varotsos et al. (2014); Coakley (2003)].

Our main objection to Björn's critique of our paper, however, is that it is based on his erroneous assumption that albedo is

the only important factor relevant to photon dissipation in the light-pigment interaction. Björn states, "Thus, it appears that if Michaelian and Simeonov are correct, one would expect organisms, in particular phototrophic organisms, or the biosphere to be less reflecting and more absorbing than dead matter." Although, as stated above, this is generally the case, one should not *a priori* "expect" this since, as we mentioned even in the abstract of our original article [Michaelian and Simeonov (2015)], albedo (reflected light) is only one part of the equation for determining global photon dissipation or entropy production which

can be attributed to pigments in life. Other factors important to entropy production are; (1) the red-shifting of the absorbed energy in the pigments, (2) dispersion of the emitted, reflected, and transmitted photon beams into a larger outgoing solid angle, and (3) the coupling of photon dissipation in pigments to other abiotic entropy producing processes, such as the water cycle [Michaelian (2012a, b)] which, i) further red-shifts the energy, and ii) allows greater proliferation and spread of organic pigments over an ever greater surface area of Earth.

Therefore, the plausibility of our assertion that the fundamental molecules of life arose as pigments and co-evolved in response to the thermodynamic imperative of dissipating the prevailing solar spectrum could only be evaluated by considering all the factors related to photon dissipation (entropy production) due to pigments, not only the albedo. This can easily be achieved by comparing the incoming solar spectrum with the outgoing global Earth reflected and emitted spectra, determining the entropy production from the differences of these spectra, and then comparing this to other similar bodies devoid of life. We

have, in fact, performed such detailed calculations of the entropy production of the Earth and compared it to that of its sister planets Venus and Mars in another paper [Michaelian (2012b)] cited in our article under discussion. We found that Earth's entropy production per unit area is almost twice that of either of its neighbors, and this may be attributed to the presence of life on Earth [Michaelian (2012b)]. Kleidon et al. (2000) have compared the surface temperatures and amount of water vapor of a simulated Earth with and without life and find 8 °C average lower temperature and 3 times the amount of water vapor

in the atmosphere for the simulation with life; the lower temperature and greater amount of water vapor implying (see below) significantly greater entropy production for a planet with life.

## 2 Albedo of the Moon, a world without life

Comparing the albedo of Earth to the Moon, in the manner Björn does to demonstrate that non-living material is better at dissipating photons than living material, is not valid because, 1) the Earth and moon are physically very different astronomical

bodies, and, more importantly, 2) albedo is a poor proxy for photon dissipation and entropy production. We can see this by performing a very simple approximate black-body calculation (see Michaelian (2012b) for a more accurate grey-body calculation) which takes into account, not only the albedo (reflected light), but also the emitted light due to the red-shift in



the pigments, emission into greater solid angle, and the additional red-shift due to photon dissipation in pigments coupling to abiotic processes such as the water cycle.

The temperature that the Earth emits into space, including the effect of the coupling of the heat of photon dissipation in pigments to the water cycle (evapotranspiration) and other irreversible processes (winds, ocean currents, convection cells, etc. ), is approximately -18 °C (255 K) corresponding to the temperature of the middle of the troposphere, at the cloud tops ($\sim$ 5 Km), averaged over latitude. Since average day and night temperatures of the middle troposphere are similar (varying by $< 0.5$ K [Muhsin et al. (2017)]) due to surface convection, winds, currents, the heat capacity of water vapor, etc., the radiation

re-emitted from Earth is effectively emitted into a $4\pi$ solid angle.

The surface temperature of the Moon during daylight hours (days are $\sim 30$ Earth days) reaches 127 °C (400 K) which is therefore the approximate black-body temperature at which the Moon surface will radiate (re-emit) into space. Since the temperature difference between day and night on the Moon is so large (127 °C to -183 °C) and since the total amount of emitted radiation goes as $T^4$ for black bodies, the majority (99.8 %) of the radiation emitted by the Moon into space occurs

during the day and thus radiated basically only into a $2\pi$ solid angle.

Not all light is absorbed by the respective surface, some of it is reflected (albedo). To simplify the calculation, we assume that the albedo is the same over all wavelengths (a wavelength dependent albedo will not change the result). Therefore, we assume that the Moon reflects 15% of all incident light and the Earth 29%. We assume, furthermore, that for both bodies this reflected light is diffuse and reflected into a $2\pi$ solid angle.

Since [Prigogine (1967)],

$$dS = dE \left( \frac{1}{T_1} - \frac{1}{T_2} \right), \tag{1}$$

the approximate ratio of the entropy production Earth/Moon per unit incident energy per unit surface area for emission plus reflection is,

$$\frac{dS_{Earth}}{dS_{Moon}} \approx \frac{0.71 \cdot \frac{4\pi}{\Omega_{Sun}} \cdot \frac{1}{(273-18)} + 0.29 \cdot \frac{2\pi}{\Omega_{Sun}} \cdot \frac{1}{5800}}{0.85 \cdot \frac{2\pi}{\Omega_{Sun}} \cdot \frac{1}{(273+127)} + 0.15 \cdot \frac{2\pi}{\Omega_{Sun}} \cdot \frac{1}{5800}} = \frac{0.0349887 + 3.1415926 \times 10^{-4}}{0.0133517 + 1.6249617 \times 10^{-4}} = \frac{0.0353028}{0.0135142} = 2.612 \tag{2}$$

where we are using a Sun surface temperature of 5800 K and $\Omega_{Sun}$ is the solid angle subtended by the Sun on Earth (or Moon), $7.193 \times 10^{-5}$ sr, but we have set it to one for the numerical values given in equation (2) since it anyway drops out of the ratio. We have also ignored the term $-1/T_2 = -1/5800$ in the first terms on the right hand side of the equation.

The entropy in the emitted plus reflected light per unit incident energy, per unit surface area, per unit time, is thus about 2.6 times greater for the Earth than for the Moon, contrary to what Björn would conclude from his albedo comparison. This

is because the black-body spectrum emitted at a lower temperature by Earth is much more red-shifted than that emitted by the Moon at a much higher temperature, and also because the Earth emits its absorbed solar energy into a $4\pi$ instead of $2\pi$ solid angle (in large part due to life increasing the amount of water vapor in the atmosphere [Kleidon (2008)]). Thus, even though the Moon absorbs more light (has lower albedo), this increase in absorption does not compensate for the greater amount of red-shift and the dispersion of the emitted light into a greater solid angle occurring on Earth attributable to pigments in life.





Simply comparing albedos of different materials, therefore, says very little about their photon dissipation potential, especially
if one of them is living material.

## 3   Vegetation compared to bare ground

Björn's second comparison of vegetation with "bare ground" suffers from the same oversight since it again considers only
albedo (reflection). Rather than looking only at reflection data, it is also important to consider the temperature data of the

emitted spectrum. There exists extensive data for this [Schneider and Kay (1994)] compiled from infrared temperature mea-
surements obtained from airplane fly-overs. The result is clear; the temperature measurements over climax ecosystems are
lower than those measured over perturbed ecosystems, and these are lower than those measured over regions devoid of life.
Another simple way of seeing this is that rocks (or ground without organic material) become much hotter under the sun than
does vegetation – albedo plays only a small part, it is the association of life with water and the water cycle that plays the greater

part.

Even beyond the red-edge, however, as mentioned above, the albedo of areas covered with vegetation is usually lower than
that devoid of life [Barry and Chorley (1992); Wang et al. (1998); Varotsos et al. (2014); Coakley (2003)].

A further note of caution is that "bare ground" is usually not devoid of life, or life produced (biological) pigments, and
water. Sufficiently developed biocrusts reduce significantly the albedo of the soils they cover [Ustin et al. (2009)]. An important

component is the cyanobacterial pigment scytonemin which reduces albedo significantly [Couradeau et al. (2016)].

## 4   The temporal aspect

The forests, as Björn correctly indicates, are sometimes buried and later burned as fossil fuel by humans. However, they
produced at least 1000 times more entropy during their lifetime than that obtained by burning the same trees as fossil fuel
today. Less than 0.1% of the free energy in sunlight goes into carbon bond making, which is how photosynthesis stores

free energy [Gates (1980)]. In living plants, more than 99.9% of solar free energy is simply turned into heat through photon
dissipation in the leaves (the dissipation involved in the process of photosynthesis itself, plus non-photochemical quenching).
This heat of dissipation is then coupled by the living system to the water cycle through evapotransporation from leaves which
increases further the photon dissipation or entropy production of Earth [Michaelian (2012b)].

The fact that a small amount of free energy available in sunlight is not instantly dissipated by ecosystems, and instead is

stored for different amounts of time, has no bearing on the point under discussion concerning whether or not pigments, life, and
ecosystems arose as a result of the thermodynamic imperative of entropy production through photon dissipation. Storage of
free energy for later use is, of course, necessary for maintaining the different trophic levels of an ecosystem, and this hierarchy
actually improves global dissipation.

Although the storage of a small amount of free energy for larger times (for example, as coal and petroleum) may make

ecosystems imperfect at dissipation, so too does; 1) the fact that their albedo is not zero, 2) that pigments absorb less strongly





at wavelengths greater than the red-edge, 3) that the physical size of the pigments are not at their theoretical limit, 4) that fluorescence reduces entropy production, i.e. the quantum efficiency for dexcitation through a conical intersection to the ground state could be further increased, 5) that pigment distribution over the whole Earth surface could be further improved. In other words, ecosystems still have room to evolve under the thermodynamic imperative towards becoming even better dissipating

systems. Modern ecosystems are, however, much more apt at dissipating sunlight than were ancient ecosystems, and this can be seen, for example, in the appearance of new pigments over time, in the spread of life over the whole Earth surface, in the fact that increases in vegetation increase water vapor in the atmosphere and this maintains day and night temperatures similar, thereby increasing the solid angle of the Earth's emitted radiation, and, in the fact that much less free energy stored in carbon bonding in organic matter is being buried today as compared to ecosystems of the past.

Nature's thermodynamic imperative of increasing global entropy production through increasing photon dissipation is indifferent to, but at the same time probably the source of, human concerns over responsibility or irresponsibility for burning fossil fuels or for preserving present ecosystems. Although most of us are blissfully unaware of it, the second law is also driving human evolution and activity. Human free energy use (dissipation) has increased exponentially over the last centuries and this trend will continue for as long as we remain a robust knowledge possessing and technical species on this planet. Our future

contribution to global dissipation would appear to go much beyond our dissipation of the chemical potential stored in fossil fuels, and beyond our traditional role as gardeners for the plants, to spreading biopigments over the whole of Earth (global greening), and even to eventually terraforming other planets.

## 5  Aquatic environments

Contrary to what Björn suggests, living organisms and free-floating, biologically-derived organic pigments – colored dissolved

organic matter, CDOM – at the ocean surface microlayer certainly do augment photon dissipation (entropy production) compared to water without organic material by, 1) increasing photon absorption at the surface, particularly for shorter wavelengths and at shallow incident photon angles, and, 2) increasing the red-shifting of the absorbed energy by coupling it to evaporation from the ocean surface microlayer (see Michaelian (2012b) and references therein). A detailed calculation of the entropy production as a function of incident photon wavelength for the ocean surface microlayer, with and without organic material, is

given section 6 of Michaelian (2012b). By absorbing and dissipating UV and visible light, the organic matter at the sea surface microlayer contributes an additional approximately 23% to the total entropy production due to photon dissipation in this layer on a clear day, and on an overcast day, it contributes an additional surprising 400% [Michaelian (2012b)]. The particular data presented by Björn on visible reflection alone, again, says little about photon dissipation or entropy production.

Forests do indeed have a tendency to increase cloudiness over land, but this is important for the other important entropy

producing process of the biosphere; the water cycle. Furthermore, cloudiness, although diffusely reflecting sunlight (also producing entropy), increases the probability of rain further inland from the coasts and thus the possibility of sustaining vegetation for photon dissipation far inland from ocean shores [Makarieva and Gorshkov (2007)]. Vegetation, by increasing the amount





of water vapor in the atmosphere [Kleidon et al. (2000)], also helps keep the temperature similar on the day and night sides of Earth, meaning emission of infrared radiation into a solid angle of $4\pi$ rather than of $2\pi$, again increasing entropy production.

A grey-body calculation of the entropy production of the Earth compared to its neighbors Mars and Venus shows that Earth's entropy production per unit surface area is almost twice that of Mars (even though, as Björn correctly points out, Mars has lower global albedo than Earth), and about 1.6 times that of Venus [Michaelian (2012b)], and this is most likely attributable to the presence of life on Earth.

## 6   Conclusions

Björn incorrectly assumes that albedo (reflection) is the only important factor related to photon dissipation in pigments giving rise to entropy production of living organisms, ecosystems, and the biosphere. He ignores the other components involved in entropy production attributable to photon interaction with biology mentioned in our original article and abstract; the shift towards the infrared of the emitted spectrum, the emission into a greater solid angle, covering all of Earth's surface with pigments, the coupling of life to other photon dissipating processes such as the water cycle. His conclusions are therefore

mistaken and do not provide legitimate reason for doubting our assertion that "we have presented evidence that supports the thermodynamic dissipation theory of the origin of life [Michaelian (2009, 2011, 2016, 2017, 2021)], which states that life arose and proliferated to carry out the thermodynamic function of dissipating the entropically most important part of the solar spectrum (the shortest wavelength photons) prevailing at Earth's surface and that this irreversible process began to evolve and couple with other irreversible abiotic processes, such as the water cycle, to become more efficient, to cover ever more

completely the electromagnetic spectrum, and to cover ever more of Earth's surface."

Finally, since our first articles, published beginning in 2005 [Michaelian (2005, 2009, 2011)], we have continued to uncover evidence pointing to a connection between photon dissipation and the origin and evolution of life. These include; 1) that many of the fundamental molecules of life strongly absorb UVC light in exactly that wavelength region that was arriving at Earth's surface during the Archean [Michaelian (2012b, a); Michaelian and Simeonov (2015); Michaelian (2016)], 2) that

many of the fundamental molecule of life possess conical intersections for rapid radiationless dissipation of the photon-induced electronic excitation energy to the ground state [Michaelian (2017, 2021)], 3) that efficient photochemical routes to production of the fundamental molecules from simple and common precursors, such as HCN and $CO_2$ in water, under UVC light have been found, and that these routes have the hallmarks of dissipative structuring [Michaelian (2017); Michaelian and Rodriguez (2019); Michaelian (2021)], 4) that we have found a DNA and RNA enzyme-less denaturing mechanism tied to UVC photon

dissipation [Michaelian and Santillán Padilla (2014); Michaelian and Santillan (2019)], 5) that the homochirality of life can be explained from the morning/afternoon temperature and UVC photon circular polarization asymmetry at the ocean surface and the temperature dependence of UVC-induced denaturing [Michaelian (2018)], 6) that the strong chemical affinity of the UVC absorbing amino acids (the aromatics), and others, to their codons and anticodons can be explained based on the thermodynamic selection of greater photon dissipation afforded to the complex [Mejía Morales and Michaelian (2020)], 7) that dissipative

structuring of the fundamental molecules under UVC light provides a simple explanation for their existence in space and on



other astronomical bodies [Michaelian and Simeonov (2017)], and, 8) that plants appear to optimize evapotranspiration (the water cycle) over photosynthesis [see Michaelian (2012a, b) and references therein].

We welcome and appreciate all challenges to our Thermodynamic Dissipation Theory for the Origin and Evolution of Life.

*Author contributions.* K. Michaelian and A. Simeonov contributed to this Reply.

*Competing interests.* The authors declare no competing interests.





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
