# Peer review of "Reply to Lars Olof Björn's Comment on our article "Fundamental molecules of life are pigments which arose and co-evolved as a response to the thermodynamic imperative of dissipating the prevailing solar spectrum""

_Biogeosciences, 2021_

## Author Response (AR1)

**Reply to Reviewer 1**

We thank the reviewer for their careful reading of our manuscript and their corrections and suggestions which have improved our manuscript. In the following, the reviewers' comments are in normal face and our replies are in bold face.

**Some of the references are not-pier reviewed or are self-published, e.g. Michaelian and Santillán Padilla 2014, and Michaelian 2009. Are there more up to date, peer reviewed articles that could replace these citations?**

We have now updated the references and included peer reviewed articles where possible.

**Section 1.**

**L19. Change: 'there are many data available' to, 'there is much data available'.**

Corrected.

**L18-20. Many forests have a high seasonal variability in their albedo. For example, in some latitudes, in wintertime, the albedo is more variable due to changing snow cover. Could the increased albedo in winter may be offsetting the higher irradiance in summer?**

Snow cover is certainly more reflective than vegetation and there is certainly some trade-off between getting enough water (snow) for plants to these dryer and colder regions and the increase of forests absorption with respect to ground without vegetation. However, an indication that nature does indeed tend to increase photon absorption and dissipation, even in colder climates at higher latitudes or altitudes, where snow is common, is that trees have evolved needles instead of leaves which are more efficient at eliminating snow cover, rapidly exposing the pigments directly to sunshine. The following picture is a good example of this.

[Figure]

**L39. What do you mean by 'bodies devoid of life'?**

We have changes this to "… and then comparing this to the same for neighboring planets which apparently do not have an evolved ecosystem."

We have further included a new paragraph making reference to our recently published paper which compares the entropy production of a leaf and a forest to that of bare ground and pure water.

**L43. How could other factors indicative of (later) life on Earth such as oxygen affect entropy?**

Through ultraviolet-induced photochemical reactions on life-produced oxygen in Earth's upper atmosphere, ozone is formed which is strongly absorptive and dissipative in the short UV wavelengths. Evidence for an oxygenated atmosphere on Earth appears 1000 to 1500 million years after the origin of life (see new references included). Throughout the evolution of life on Earth, new pigments covering ever more of the solar spectrum have come into being, thereby increasing global entropy production. These more recent pigments are also dissipative structures, but now dissipatively structured through complex biosynthetic pathways using visible light. Mention of this is now included in Section "Ancient Life" of the new version of our Reply.

**Section 3**

**L94. Reverse 'does vegetation'. It should read 'vegetation does'.**

Corrected.

**L94-94. Further explain the role of water and the water cycle.**

By expending free energy to convert liquid water in the leaf into an atmospheric gas, which then releases far infrared photons when it condenses at the cold cloud tops, the water cycle increases further still photon dissipation. This is included in the new version of our Reply.

**L96. 'Even beyond the red-edge.' This does not follow on from the section above.**

This has been removed.

**L99. Reverse 'significantly' and 'reduce'. 'Significantly reduce'.**

Corrected.

**L99-100. 'An important component within biocrusts is the cyanobacterial pigment scytonemin which significantly reduces albedo...'**

Corrected.

**Section 4.**

**L104. Include a reference at the end of the first sentence.**

The sentence has been restructured such that the quoted value of 1000 is now more obviously related to the value of 0.1% of free energy which goes into carbon bonding of photosynthesis. The sentence now reads "However, during their lifetime trees produce at least 1000 times more entropy than obtained by being burnt as fossil fuel today since less than 0.1\% of the free energy in sunlight goes into carbon bond making, which is how photosynthesis stores free energy [Gates1980]."

**L107-8. "...which further increases the photon dissipation....'**

Corrected.

**L114. What is meant by 'larger times'?**

This has been changed to "Although the storage of a very small amount of free energy in a practically inaccessible form (for example, deposited underground as coal or petroleum) ,,,"

**L120-124. This sentence is too long, it needs restructuring.**

We have restructured this sentence as a list of points.

**L128-132. I am not sure I follow the logic of this section. Please explain or expand on this to make it understandable.**

We have rewritten this paragraph, shortening it, hopefully making it more understandable, and including a reference. It now reads;

"Although most of us are blissfully unaware of it, the second law is also driving human evolution and activity. Human free energy use (dissipation) has increased exponentially over the last few centuries and this trend is projected to continue for as long as we remain a knowledge possessing species. Our future contribution to global dissipation will almost certainly go much beyond our present dissipation of the chemical potential stored in fossil fuels, or beyond our animal role as gardeners (e.g., fertilizers and seed spreaders) for the photon dissipating plants. We have already significantly increased the entropy production of Earth through global greening [PiaoEtAl2020] and look soon to be set for terra-forming other planets. "

**Section 5.**

**L144. Do you mean 'forests'?**

This paragraph was taken out here and inserted as a reduced version in Section "Present vegetation compared to bare ground"

**Further Corrections:**

We have improved the redaction throughout the text and removed the section concerning the entropy production of the Moon, since this section was left out of the final version of Bjorn's Comment.

**Reply to Reviewer 2**

We thank the reviewer for their interesting questions and for their useful suggestions for our Reply.

**The authors reply to Lars Olof Björn's comments on their article "Fundamental molecules of life are pigments which arose and co-evolved as a response to the thermodynamic imperative of dissipating the prevailing solar spectrum". They highlight Björn's critique points and respond in a detailed manner to his comments. I find the idea fascinating and intriguing and enjoyed reading it.**

**You mention that earth reflects 29% of incident light (line 69), then proceed to calculate the entropy ratio production (equation 2). The results suggest a 2.6 times greater entropy production for earth due to the life on earth (Line 84-85). I was curious, do you expect the ratio to be close to 1, if you use parameters from an earth without life (Line 44)? Perhaps such calculations might strengthen your assertion, if possible, to make that calculation.**

The value of 2.6 times greater entropy production that the reviewer refers to in our Reply is for the Earth/Moon comparison. The Moon is quite different from an Earth without life, since the Moon has no atmosphere and therefore no water cycle, or any other irreversible process occurring at its surface. The Moon thus radiates as an approximate black-body at the temperature of its surface as determined by energy conservation, taking into account its albedo, emissivity, and distance from the Sun. The Moon's entropy production can thus be easily calculated and compared to that of the Earth today.

However, an Earth without life would have an atmosphere, probably more similar to that of Titan than to Earth's non-equilibrium atmosphere of today which has been completely transformed by life.  An Earth without life would have little oxygen, little ozone, and therefore intense UV light at the surface. Without life, Earth would probably have lost most of its water since there would be no free oxygen to capture hydrogen freed by UV photolysis. Surface temperatures would be different (probably cooler because of less greenhouse water vapor) and this would affect the albedo through increasing ice at the polar caps. Life is also thought to have had an important effect on plate tectonics (Parnell, and Brolly, 2021), etc. In summary, a fair comparison of the entropy production of an Earth with and without life would require knowledge of an alternative geochemical history of Earth, as well as knowledge of what irreversible processes could be occurring on such an Earth today without life.

Kleidon et al. (2000) consider a simple, and imaginary, scenario in which life is suddenly removed while maintaining the atmosphere of today relatively unchanged except for the amount of water vapor. This is obviously not representative of what would be the case if life had failed to arise 3.9 billion years ago, and it's not what the reviewer is asking. However, for this scenario they conclude that, "Land surface evapotranspiration more than triples in the presence of the 'green planet', land precipitation doubles (as a second order effect) and near surface temperatures are lower by as much as 8 K in the seasonal mean resulting from the increase in latent heat flux." Higher surface temperatures emitting into space (without the latent heat flux) and less damping of the day-night temperature difference (due to less water vapor) for an Earth without life would mean less entropy production, as we found for the exaggerated case of the Moon.

The ratio of the entropy production of an Earth with life over an Earth without life, would most probably be less than the value of 2.6 found for comparison with the Moon (because an Earth without life would still have secondary dissipative processes occurring at its surface), perhaps similar to our comparison with Venus, Earth/Venus of 1.247 W $K^{-1}$ $m^{-2}$ / 0.801 W $K^{-1}$ $m^{-2}$ = 1.557 (Michaelian 2012). However, there is no conceivable way of obtaining a reliable answer to the reviewer's question without knowledge of the alternative geochemical history.

An incomplete, but more meaningful, answer would be to consider just that part of the entropy production due to absorption and dissipation of incident photons on surfaces with and without life, without considering secondary coupled irreversible process. In Michaelian and Cano (2022) we performed a detailed calculation for the entropy production of a conifer forest and compared it to that of a dry sand and rock desert. The ratio of the entropy production we found, with the sun directly overhead, was, 6.834 W $K^{-1}$ $m^{-2}$ /4.714 W $K^{-1}$ $m^{-2}$ = 1.450

Since Björn removed his section on the comparison of the albedo of the Moon to that of the Earth from his final published Comment, we have also left this section out in our revised version of our Reply.

**Comments:**

**Line 35: I am curious if there are any examples of which types/specific enzymes might be the first to arise from the fundamental molecules based on your assertions?**

We have published a paper (Mejia and Michaelian, 2020) on how UV-C photon dissipation could have led to the specificity (chemical affinity) of amino acids for their codons and anticodons, indicative of a stereochemical era at the beginning of life (Yarus,1998; Yarus et al., 2005). It seems that those amino acids which one could argue have some capacity to increase photon dissipation of the amino acid - nucleic acid complex (e.g., amino acid acting as a UV-C antenna, as an aliphatic anchor to the ocean surface, or as a facilitator for charge transfer catalysis) are

precisely those that have strong chemical affinity to their codons or anticodons. We explain in our paper how this may have come about through a thermodynamic selection on photon dissipation. As an example, the aromatic amino acids tyrosine, tryptophan, phenylalanine, and histidine could have acted as antenna molecules when coupled to the nucleic acids (which have conical intersections for rapid internal conversion to the ground state) giving the complex more efficient UV-C dissipation. If reproduction (e.g. denaturing) depended on photon dissipation (for example, through local heating or photon-induced charge transfer - Michaelian and Santillan, 2019) then those DNA or RNA sequences with chemical affinity to the antenna amino acids would be selected more frequently (because they denatured more frequently) than arbitrary sequences.

Bioinformatic studies have shown that enzymes involved in either purine metabolism or porphyrin and chlorophyll metabolism are evolutionarily the oldest (Caetano-Anolles et al. 2008, Caetano-Anolles et al. 2012). Both groups use nucleotides as their cofactors. The second group is specifically involved in pigment metabolism.

**Line 102-104: "However, they produce……today". Needs a reference.**

The sentence has been reworked so that the value of 1000 for the entropy produced during their lifetime compared with the entropy that could be produced through the burning of the tree as fossil fuel can be understood to come from the value of 0.1% of the solar free energy captured by the tree that goes into covalent bonding (Gates, 1980).

**Line 120-124: A reference is needed**

 Included.

**Some words are unnecessary negatively targeting Björn. To keep a proper and neutral discussion, either leave the words out or tone it down. Examples:**

**Line 6; "erroneous"**

**Line: 11: "…incorrect and his analysis does not provide legitimate reason for doubting…"**

**Line 24: "erroneous"**

**Line 155: "Björn incorrectly assumes…"**

**Line 159-160: "His conclusions are therefore mistaken and do not provide legitimate reason for doubting…"**

The negativity has been removed from our revised Reply.

**References:**

Caetano-Anollés G, Yafremava LS, Gee H, Caetano-Anollés D, Kim HS, Mittenthal JE. The origin and evolution of modern metabolism. Int J Biochem Cell Biol. 2009 Feb;41(2):285-97. doi: 10.1016/j.biocel.2008.08.022. Epub 2008 Aug 26. PMID: 18790074.

Caetano-Anollés G, Kim KM, Caetano-Anollés D. The phylogenomic roots of modern biochemistry: origins of proteins, cofactors and protein biosynthesis. J Mol Evol. 2012 Feb;74(1-2):1-34. doi: 10.1007/s00239-011-9480-1. Epub 2012 Jan 1. Erratum in: J Mol Evol. 2012 Feb;74(1-2):35-6. PMID: 22210458.

Gates, D. M, Biophysical Ecology, Springer-Verlag, 1980, ISBN0-387-90414-X.

Kleidon, A., Fraedrich, K., and Heimann, M. A.: Green Planet Versus a Desert World: Estimating the Maximum Effect of Vegetation on the Land Surface Climate, Climatic Change, 44, 471–493, https://doi.org/10.1023/A:1005559518889, 2000.

Mejia, J. and Michaelian, K., Photon Dissipation as the Origin of Information Encoding in RNA and DNA. Entropy 2020, 22 (9), 940. https://www.mdpi.com/1099-4300/22/9/940

Michaelian, K.: Biological catalysis of the hydrological cycle: life's thermodynamic function, Hydrol. Earth Syst. Sci., 16, 2629–2645, https://doi.org/10.5194/hess-16-2629-2012, 2012.

Michaelian, K. and Santillán Padilla, N. UVC photon-induced denaturing of DNA: A possible dissipative route to Archean enzyme-less replication, Heliyon 5, e01902 (2019). https://www.heliyon.com/article/e01902

Michaelian, K., Cano Mateo, R.E. A Photon Force and Flow for Dissipative Structuring: Application to Pigments, Plants and Ecosystems. Entropy 2022, 24, 76. https://doi.org/10.3390/e240

Parnell, J., Brolly, C. Increased biomass and carbon burial 2 billion years ago triggered mountain building. *Commun Earth Environ* **2,** 238 (2021). https://doi.org/10.1038/s43247-021-00313-5

Yarus, M. Amino Acids as RNA Ligands: A Direct-RNA-Template Theory for the Codes Origin. J. Mol. Evol. 1998, 47, 109–117.

Yarus, M.; Caporaso, J.G.; Knight, R. Origins The Genetic Code: The Escaped Triplet Theory. Annu. Rev. Biochem. 2005, 74, 179–198.